# The impact of heating, ventilation and air conditioning (HVAC) design features on the transmission of viruses, including the 2019 novel coronavirus (COVID-19): A systematic review of humidity

Gail M. Thornton[ORCID][1], Brian A. Fleck[ORCID][1]*, Dhyey Dandnayak[ORCID][1], Emily Kroeker[1], Lexuan Zhong[ORCID][1], Lisa Hartling[ORCID][2]

1 Department of Mechanical Engineering, Faculty of Engineering, University of Alberta, Edmonton, Canada,
2 Alberta Research Centre for Health Evidence, Department of Pediatrics, Faculty of Medicine & Dentistry, University of Alberta, Edmonton, Canada

* brian.fleck@ualberta.ca

## Abstract

The aerosol route has been a pathway for transmission of many viruses. Similarly, recent evidence has determined aerosol transmission for SARS-CoV-2 to be significant. Consequently, public health officials and professionals have sought data regarding the role of Heating, Ventilation, and Air Conditioning (HVAC) features as a means to mitigate transmission of viruses, particularly coronaviruses. Using international standards, a systematic review was conducted to comprehensively identify and synthesize research examining the effect of humidity on transmission of coronaviruses and influenza. The results from 24 relevant studies showed that: increasing from mid (40–60%) to high (>60%) relative humidity (RH) for SARS-CoV-2 was associated with decreased virus survival; although SARS-CoV-2 results appear consistent, coronaviruses do not all behave the same; increasing from low (<40%) to mid RH for influenza was associated with decreased persistence, infectivity, viability, and survival, however effects of increased humidity from mid to high for influenza were not consistent; and medium, temperature, and exposure time were associated with inconsistency in results for both coronaviruses and influenza. Adapting humidity to mitigate virus transmission is complex. When controlling humidity as an HVAC feature, practitioners should take into account virus type and temperature. Future research should also consider the impact of exposure time, temperature, and medium when designing experiments, while also working towards more standardized testing procedures.

**Clinical trial registration:** PROSPERO 2020 CRD42020193968.

## Introduction

The World Health Organization (WHO) declared, in March 2020, a global pandemic due to Coronavirus Disease 2019 (COVID-19) which is caused by Severe Acute Respiratory Syndrome coronavirus 2 (SARS-CoV-2) [1, 2]. Throughout the world, public health authorities

**Data Availability Statement:** All relevant data are within the paper and its Supporting Information files.

**Funding:** This work is funded by a Canadian Institutes of Health Research (CIHR) Operating Grant, Canadian 2019 Novel Coronavirus (COVID-19) Rapid Research Funding Opportunity [https://webapps.cihr-irsc.gc.ca/decisions/p/project_details.html?applId=422567&lang=en], and Alberta Innovates. Dr. Hartling is supported by a Canada Research Chair in Knowledge Synthesis and Translation. Drs. Fleck and Zhong are supported by the NSERC Discovery program. The funders played no role in study design, data collection and analysis, decision to publish, or preparation of the manuscript.

**Competing interests:** The authors have declared that no competing interests exist.

have sought evidence regarding virus transmission routes and corresponding public health measures to mitigate virus spread. Certain viruses can be transmitted via an aerosol route [3], facilitated by virus-laden aerosols, which are expelled by humans, that remain airborne for extended periods of time. Recent evidence suggests that, particularly in indoor environments with poor ventilation, SARS-CoV-2 can spread via airborne transmission [4, 5]. The American Society of Heating, Refrigerating, and Air-Conditioning Engineers (ASHRAE) released a statement in April 2021 declaring that "airborne transmission of SARS-CoV-2 is significant and should be controlled. Changes to building operations, including the operation of heating, ventilating, and air-conditioning systems, can reduce airborne exposures" [6]. As a result, determining the appropriate measures to help protect occupants of indoor spaces based on informed, interdisciplinary research is critical to managing and controlling the spread of infectious disease [7]. Heating, ventilation, and air conditioning (HVAC) systems can be used to mitigate airborne transmission of viruses by diluting or removing the contaminated air where humans breathe from inside the building envelope [7–10]. HVAC design features, particularly humidity, can influence transmission.

As part of their 2021 recommendations for minimizing infectious aerosol exposure, ASHRAE recommended "maintaining temperatures and humidity at set points," potentially highlighting the role of humidity in transmission [11]. Previous systematic reviews have also noted the impact of humidity on infectious agents [4, 12, 13]. Derby et al. reviewed the effect of low humidity (≤40% relative humidity [RH]) on virus viability and transmission [12] and identified several studies, both modelling and experimental, showing that humidity influenced virus transmission and virus survival [14–18]. Some of these studies found that increasing humidity from low RH levels to approximately 50%RH was associated with decreased transmission [14, 15, 18]. Other reviews have also highlighted the effect of temperature, exposure time, and air sampling techniques [4, 12, 13]. While Derby et al. [12] focused on the impact of low humidity levels (<40%RH), they also grouped humidity levels to allow for comparisons across studies: low (20–30%RH), mid (~50%RH), and high (70–90%RH).

Coronaviruses have emerged as infectious agents of great concern for potential airborne transmission. Coronaviruses are lipid enveloped, single-stranded RNA (ssRNA) viruses [19]. Seven human coronaviruses have been identified; however, SARS-CoV-2, Severe Acute Respiratory Syndrome coronavirus (SARS-CoV), and Middle East Respiratory Syndrome coronavirus (MERS-CoV) have received the most attention due to their pathogenicity and lethality [20]. These coronaviruses had their first emergence in the last 18 years [20], with SARS-CoV in 2003, MERS-CoV in 2012, and SARS-CoV-2 in 2019. However, due to the potentially limited number of studies examining coronaviruses, studies examining the influence of humidity on influenza viruses may also provide useful information. As virus envelopes were found to be an important factor in virus transmission [12, 13], Influenza (both A and B strains) was chosen for inclusion in the present review due to its structure as a lipid enveloped, ssRNA virus [14].

As mentioned, previous reviews have studied the role of humidity in virus transmission in some capacity [4, 12, 13]. This systematic review builds on these previous reviews through an extensive and comprehensive search of the literature to identify and synthesize published research determining the impact of humidity in reducing virus transmission. While Derby et al. [12] focused on the role of low humidity, the present review seeks to provide a broader picture including all humidity levels. As well, this review focuses on the enveloped, ssRNA coronaviruses and influenza viruses as opposed to viruses more generally. By doing so, the insight drawn from this review could help answer questions of the role of humidity in SARS-CoV-2 transmission mitigation in mechanically ventilated indoor environments. As well, a detailed examination of the existing scientific literature can identify gaps in current research, which can guide future research priorities.

## Methods

As part of a larger research program to review the literature on HVAC design feature and airborne virus transmission, this systematic review was performed to identify and synthesize the scientific literature regarding the impact of humidity on virus transmission within the built environment. Results for other design features of interest (ventilation, ultraviolet radiation, and filtration) are reported separately. The systematic review is registered (CRD42020193968) and a protocol was developed a priori and made publicly available [21, 22]. Standards, as defined by the international Cochrane organization [23], for the conduct of systematic reviews were followed with modifications for questions related to etiology [24]. Additionally, the review was reported according to relevant reporting standards [25].

### Search strategy

Using concepts related to virus, transmission, and HVAC, a research librarian (GMT) searched three electronic databases (Ovid MEDLINE, Compendex, Web of Science Core) from inception to June 2020 (see Appendix A in S1 File for the Ovid MEDLINE search strategy). Prior to implementing the searches, two librarians peer-reviewed the strategies (TL, AH). An updated search was conducted in January 2021. Reference lists of all relevant papers and review articles were screened. Using Compendex and Web of Science, conference abstracts were identified and were not included, but literature was searched to identify if any relevant abstracts had been published as complete papers. Limits for language or year of publication were not placed on the search. However, only English-language studies were included due to the volume of available literature and resource constraints. References were managed in End-Note and duplicate records removed prior to screening.

### Study selection

Study selection occurred in two stages: title/abstract screening and full-text screening. In the first stage, two reviewers independently screened the titles and abstracts of all references identified by the searches of the electronic databases. Relevance of each record was classified as No, Yes, or Maybe. Conflicts between No and Yes/Maybe were resolved by one of the review team. Pilot testing was conducted with three sets of studies (n = 199 each) to develop consistency among the review team. The review team met to discuss discrepancies and develop decision rules after each set of pilot screenings. In the second stage, two reviewers independently reviewed the full-text articles and applied the inclusion/exclusion criteria. Reviewers classified studies as Exclude or Include. Conflicts between Exclude or Include were resolved through consensus by the review team. One reviewer resolved conflicts when different exclusion reasons were given. Pilot tests with three sets of studies (n = 30 each) were used for the second stage of screening. The review team met to resolve discrepancies after each pilot round. Covidence software was used to conduct screening.

### Inclusion and exclusion criteria

Exclusion and inclusion criteria are listed in Appendix B in S1 File. This systematic review was part of a larger effort to examine virus transmission and different HVAC design features. While all four design features were included in the search and screening process, only studies evaluating humidity were synthesized here. In addition, literature examining humidity in combination with ultraviolet radiation was addressed in a separate systematic review on ultraviolet radiation. A variety of agents were included in the search with priority placed on studies of viruses or agents that simulated viruses. Other agents (e.g., fungi, bacteria) would be included

if studies were not available specific to viruses. Studies using bacteriophages, which are viruses that infect bacterial cells [26], were included. For this specific review, the synthesis was further narrowed from viruses to coronaviruses and influenza viruses. Studies of the indoor built environment (e.g., office, public, residential buildings) which had mechanical ventilation were of particular interest. Primary research providing quantitative results of the association between humidity and virus transmission was included. Only English-language, peer-reviewed publications were included.

## Risk of bias assessment

For experimental studies, the risk of bias was determined based on three key domains: selection bias, information bias and confounding [27, 28]. Reviewers assessed domains as high, low, or unclear risk of bias using signalling questions [23] for the different study types that were included (e.g., animal studies, laboratory experiments, epidemiological studies) from guidance documents [27–30]. Modelling studies were assessed using the following three key domains: definition, assumption, and validation [30, 31]. Definition considered model complexity and data sources, assumption considered the explanation and description of model assumptions, and validation considered model validation and sensitivity analysis [31]. Reviewers assessed each domain as high, low, or unclear risk of bias based on signalling questions [30–32]. Pilot tests were conducted among three review authors for risk of bias items, then two reviewers (DD, EK, or NF) applied the criteria to each relevant study independently and met to resolve discrepancies.

## Data extraction

General information about the study (authors, year of publication, country of corresponding author, year of publication, study design) and methods (setting, population [as applicable], intervention set-up, agent studied) was extracted. Details on humidity treatment parameters (where available) were extracted, including relative humidity (RH), absolute humidity (AH), medium, exposure time, and temperature, where applicable. The studies were grouped as aerosolized virus, modelling, animal, and field studies. Quantitative data were extracted, in addition to the results of any tests of statistical significance related to humidity. The primary outcome of interest was quantitative measures of the association between virus transmission and humidity. As such, data on actual transmission were extracted where available (i.e., infections), as well as information regarding virus survival, persistence, infectivity, viral load per hour, concentration, recovery, decay rate, death rate, and virus detection in air. In the animal and aerosolized virus tables, humidity was categorized as low (<40%RH), mid (40–60%RH), and high (>60%RH) RH. Symbols were used to denote high virus survival (+), low virus survival (-), mid virus survival (/) (i.e., between low and high), and no effect (*). The abbreviation ND was used when virus was not detected, and NR was used when the virus viability in a particular humidity category was not reported. One reviewer extracted data and a second reviewer verified data for accuracy and completeness using a data extraction form spreadsheet to ensure consistency. The review team discussed discrepancies.

## Data synthesis

Meta-analysis was not possible due to heterogeneity across studies in terms of study design, humidity levels tested, outcomes assessed, and results reported. Evidence tables were developed to describe the studies and their results. A narrative synthesis of results was conducted by study grouping (aerosolized virus, modelling, animal, and field studies).

## Results

The searches yielded 12,177 unique citations. 2,428 were identified as potentially relevant based on title/abstract screening and 568 met the inclusion criteria (Fig 1). 124 studies were relevant to humidity with 65 relevant to viruses more broadly. Of those 65, 24 were specific to lipid enveloped, ssRNA viruses: coronavirus (n = 6) and influenza (n = 18). Two relevant studies [33, 34] were related and are considered as one in the analyses that follow, therefore, 23 studies were synthesized. Studies were published between 1943 and 2020 (median year 2013). The majority of studies (n = 10) were laboratory experiments, with six experimental animal studies, one field observational study, and six modelling studies. Details of individual studies are provided in tables and summarized in the sections that follow; humidity was categorized as low (<40%RH), mid (40–60%RH), and high (>60%RH) RH. Studies were funded by national research funding organizations (n = 15) and public foundations (n = 2), with three studies reporting no external funding and three studies not reporting funding sources.

### Aerosolized viruses

**Coronaviruses.** Five experimental studies examined coronaviruses using SARS-CoV-2 (BetaCoV/USA/WA1/2020) [35], SARS-CoV-2 England-2 [36], MERS-CoV isolate HCoV-EMC/2012 [37, 38], and hCoV-229E [39] (Table 1). These studies were conducted by aerosolizing the virus into a rotating drum [35, 36, 38, 39] or environmental chamber [37].

Two studies that examined SARS-CoV-2 showed different results. Smither et al. [36] found that increased humidity from mid to high RH was associated with increased survival in both artificial saliva (AS) and tissue culture medium (TCM), although survival in TCM was less at higher RH than in AS at the same RH. Schuit et al. [35] found that humidity alone did not significantly affect virus survival. Discrepancy in results could be due to differences in the studies' experimental set-up and test procedures, e.g., exposure time up to 60 [35] vs 90 [36] minutes.

Two studies analyzed the effect humidity on MERS viruses, including MERS-CoV [37] and MERS-CoV isolate HCoV-EMC/2012 [38]. Van Doremalen et al. [37] found increased humidity from mid to high RH was associated with decreased virus survival (i.e., the highest survival was at mid RH), noting a significant effect of humidity on virus survival. Van Doremalen et al. [37] did not test at low RH. Pyankov et al. [38] found that increasing RH was associated with increasing virus survival when coupled with decreasing temperature; these results were statistically significant during the 30- and 60-minute exposure times. Mid RH levels were not tested.

Ijaz et al. [39] examined a full spectrum of RH ranges and found that increasing humidity from low to mid RH was associated with increased virus survival for hCoV-229E, with the highest survival for coronavirus at mid RH. As well, increased humidity from mid to high RH was associated with decreased hCoV-229E survival.

Aerosolized coronaviruses were not consistent as to minimum and maximum survival versus humidity. Two studies found that high humidity was associated with minimum virus survival for MERS-CoV [37] and hCoV-229E at 20±1˚C [39] (70%RH and 80±5%RH, respectively). Two studies found low humidity was associated with minimum survival for MERS-CoV [38] and hCoV-229E at 6±1˚C [39] (24%RH/38˚C and 30±5%RH, respectively). One study found that mid RH was associated with minimum virus survival [36]. Two studies found that mid RH was associated with maximum virus survival (50±5%RH and 40%RH) [37, 39] and two studies found that high RH was associated with maximum virus survival (79%RH/25˚C and 68–88%RH) [36, 38]. Schuit et al. [35] did not find a significant effect of humidity so minimum and maximum survival could not be determined.

**Influenza.** Six studies analyzed the effect of humidity on influenza viruses (Table 1) [16, 18, 33, 34, 37, 40, 41]. Influenza strains investigated included Influenza A PR8 [18, 33, 34],

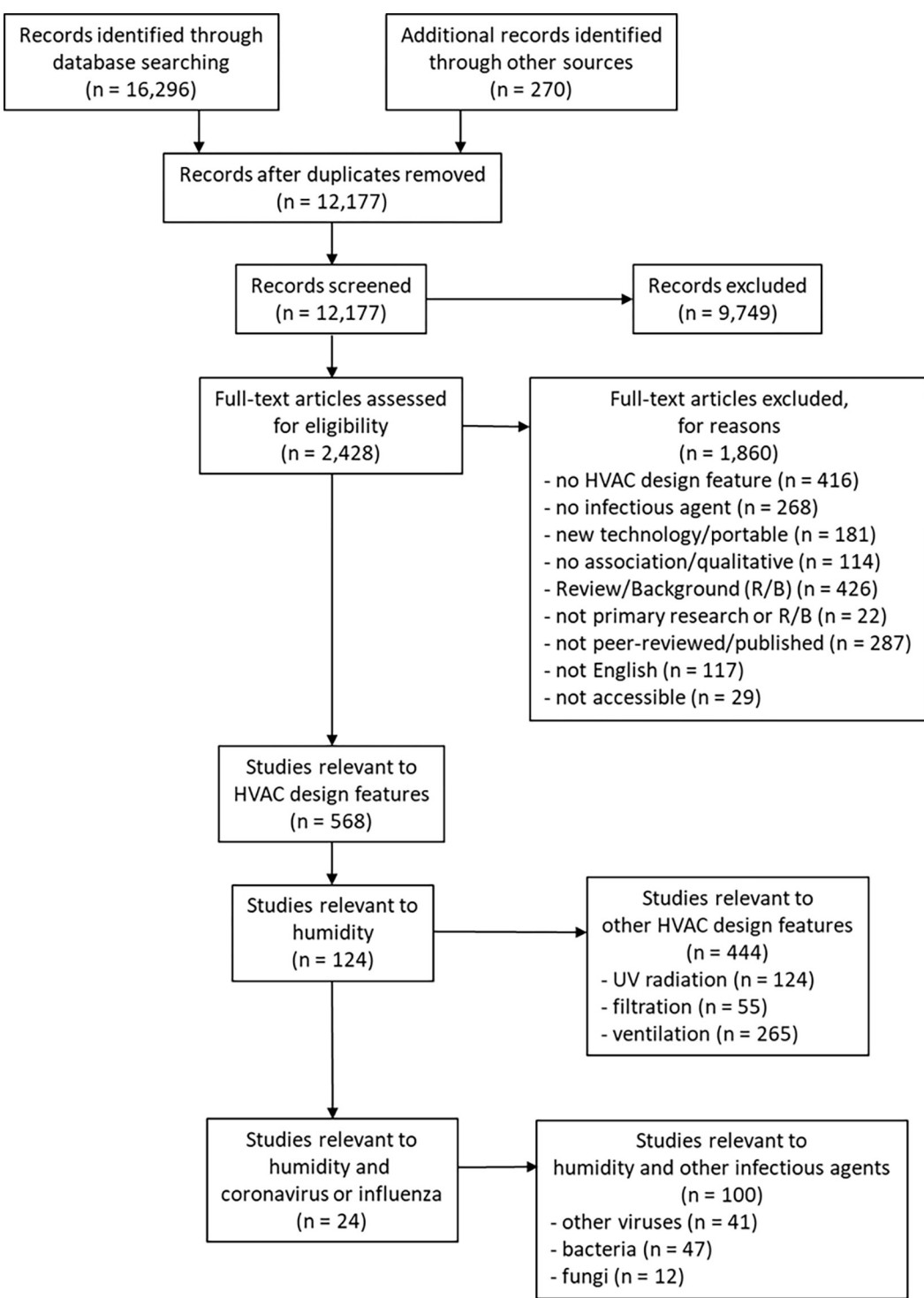

**Fig 1. Flow of studies through the selection process.** (note: search was conducted for all HVAC design features but only studies of relative humidity and coronavirus or influenza are included in this manuscript).

**Table 1. Aerosolized virus.**

| First author Year Country | Experimental design | Outcome | Virus | Effect of Humidity | | | Medium | Exposure Times | Temperature | Result | Association |
|---|---|---|---|---|---|---|---|---|---|---|---|
| | | | | Low | Mid | High | | | | | |
| **Coronaviruses** | | | | | | | | | | | |
| Ijaz 1985 [39] Canada | Coronavirus 229E was aerosolized into a rotating drum | Humidity vs recovery | hCoV-229E | / | + | - | Tryptose Phosphate Broth | 7 min, 24, and 72 hrs | 20 ± 1˚C | Increased RH from 30 ± 5% to 50 ± 5% associated with increased recovery (hCoV-229E half life from 26.76 ± 6.21 h to 67.33 ± 8.24 h) Increased RH from 50 ± 5% to 80 ± 5% associated with decreased recovery (hCoV-229E half life from 67.33 ± 8.24 h to 3.34 ± 0.16 h) | Increased RH from low to mid levels associated with increased recovery Increased RH from mid to high levels associated with decreased recovery Minimum recovery was associated with high RH (80 ± 5% RH) at 20 ± 1˚C Minimum recovery was associated with low RH (30 ± 5% RH) at 6 ± 1˚C Maximum recovery was associated with mid RH (50 ± 5% RH) at 6 ± 1˚C and 20 ± 1˚C |
| | | | | - | + | / | | | 6 ± 1˚C | Increased RH from 30 ± 5% to 50 ± 5% associated with increased recovery (hCoV-229E half life from 34.46 ± 3.21 h to 102.53 ± 9.38 h) Increased RH from 50 ± 5% to 80 ± 5% associated with decreased recovery (hCoV-229E half life from 102.53 ± 9.38 h to 86.01 ± 5.28 h) | |
| Van Doremalen 2013 [37] USA | MERS-CoV was aerosolized in an environmental chamber | Humidity vs viability | MERS (MERS-CoV isolate HcoV-EMC/2012) | NR | + | - | Dulbecco's Modified Eagle Medium | Continuous sampling during aerosolizati-on | 20˚C | Increased RH from 40% to 70% associated with *significant* decrease in MERS-CoV viability | Increased RH from mid to high levels associated with decreased viability Low RH was not reported Minimum viability associated with high RH (70%RH) Maximum viability associated with mid RH (40%RH) |
| Pyankov 2018 [38] Russia | MERS (MERS-CoV isolate HcoV-EMC/2012) was aerosolized into a rotating drum | Humidity vs virus survival | MERS (MERS-CoV isolate HCoV-EMC/2012) | - | NR | + | Dulbecco's Modified Eagle Medium supplemented with 2% fetal calf serum | 0, 15, 30, and 60 min | 79% RH and 25˚C vs 24% RH and 38˚C | Increasing RH from 24% (38˚C) to 79% (25˚C) associated with increased virus survival | Increasing RH from low (24% RH/38˚C) to high (79% RH/25˚C) levels associated with increased virus survival Mid RH levels not reported Minimum survival associated with low RH (24%/38˚C) Maximum survival associated with 79% (79%/25˚C) |
| Smither 2020 [36] United Kingdom | SARS-CoV-2 England-2 was aerosolized into a rotating drum | Humidity vs virus survival | SARS-CoV-2 England-2 | NR | - | + | Tissue Culture Medium (TCM) | 0, 15, 30, 60, and 90 min | 19–22˚C | Increased RH from 40–60% to 68–88% associated with increased survival of SARS-CoV-2 England-2 | Increased RH from mid to high levels associated with increased survival in TCM at all sample times Low levels not reported Minimum survival associated with mid RH (40–60%RH) Maximum survival associated with high RH (68–88%RH) |
| | | | | NR | - | + | Artificial Saliva (AS) | | | Increased RH from 40–60% to 68–88% associated with little difference in survival of SARS-CoV-2 England-2 at 0 minutes. Increased RH from 40–60% to 68–88% associated with increased survival at 15 minutes Increased RH from 40–60% to 68–88% associated with slightly increased survival of SARS-CoV-2 England-2 at 30 minutes increased RH from 40–60% to 68–88% associated with relatively no difference in survival at 60 minutes Increased RH from 40–60% to 68–88% associated with increased survival at 90 minutes | 0 minutes: Increased RH from mid to high levels associated with little change in AS 15, 30, and 90 minutes: Increased RH from mid to high levels associated with increased survival in AS 60 minutes: Increased RH from mid to high levels associated with little change in AS Minimum survival associated with mid RH (40–60%RH) Maximum survival associated with high RH (68–88%RH) |
| Schuit 2020 [35] USA | SARS-CoV-2 (BetaCoV/USA/WA1/2020) was aerosolized into a rotating drum | Humidity vs decay rate | SARS-CoV-2 (BetaCoV/USA/WA1/2020) | * | * | * | Simulated saliva or fresh culture medium | 30 sec, every 5 min up to 1 hr | 20˚C | Increased RH *did not significantly* affect decay rate of deSARS-CoV-2 (Covid 19) for samples taken up to 60 minutes. "While a similar effect was not observed for SARS-CoV-2 in the present study, it is possible that the shorter test durations used in the present study precluded detection of this effect of relative humidity. It is possible that additional tests of longer duration without simulated sunlight would allow a better assessment of the effect of relative humidity on SARS-CoV-2 in aerosols" (p.568) | Increased RH had *no significant effect* on decay rate RH % for minimum and maximum decay not determined |
| **Influenza** | | | | | | | | | | | |

*(Continued)*

**Table 1.** (Continued)

| First author Year Country | Experimental design | Outcome | Virus | Effect of Humidity | | | Medium | Exposure Times | Temperature | Result | Association |
|---|---|---|---|---|---|---|---|---|---|---|---|
| | | | | Low | Mid | High | | | | | |
| Hemmes 1960 [18] Netherlands | Influenza A was aerosolized in a 4 m³ test room | Humidity vs death rate and virus survival | Influenza A virus (PR₈) | + | / | - | allantoic fluid and 2% Difco peptone | "adequate" intervals of time | 20˚C | Increased RH from ~15% to ~90% associated with an increased death rate of influenza virus and a sharp transition between 40–60% RH and another sharp transition at 80% RH | Increased RH from low to mid levels associated with increased death rate (decreased survival) Increased RH from mid to high levels associated with increased death rate (decreased survival) Minimum survival associated with high RH (~90%RH) Maximum survival associated with low RH (~15%RH) |
| Harper 1961, 1963 [33, 34] England | Influenza A was generated into a rotating drum | Humidity vs viability | Influenza A | + | / | - | Allantoic fluid | 0, 0.1, 0.5,1, 4, 6, 23 hrs. | 7–8˚C | Low to mid RH (& low to high RH): Increased RH from low to mid levels associated with decreased viability at 0, 0.1, 4, 6, 23 h; Increased RH from low to mid levels associated with similar viability at 0.5, 1 h; Mid to high RH: Increased RH from mid to high levels associated with increased viability at 0, 0.1 h; Increased RH from mid to high levels associated with similar viability at 0.5, 1, 4, 6 h; Increased RH from mid to high levels associated with decreased viability at 23 h | Low to mid (& low to high RH) 0, 0.1, 4, 6, 23 h: Increased RH from low to mid levels associated with decreased viability 0.5, 1 h: Increased RH from low to mid levels associated with similar viability Mid to high RH 0, 0.1 h: Increased RH from mid to high levels associated with increased viability 0.5, 1, 4, 6 h: Increased RH from mid to high levels associated with similar viability 23h: Increased RH from mid to high levels associated with decreased viability Minimum viability associated with high RH Maximum viability associated with low RH |
| | | | | + | / | - | | | 20.5–24˚C | Low to mid RH (& low to high RH): Increased RH from low to mid levels associated with decreased viability at 0.1, 0.5, 1, 4, 6, 23 h; Increased RH from low to mid levels associated with similar viability at 0 h Mid to high RH: Increased RH from mid to high levels associated with similar viability (see Harper 1963) | Low to mid RH (& low to high RH): Increased RH from low to mid levels associated with decreased viability at 0.1, 0.5, 1, 4, 6, 23 h; Increased RH from low to mid levels associated with similar viability at 0 h Mid to high RH: Increased RH from mid to high levels associated with similar viability (see Harper 1963) Minimum viability associated with mid and high RH Maximum viability associated with low RH |
| | | | | + | / | - | | | 32˚C | Low to mid RH (& low to high RH): Increased RH from low to mid levels associated with decreased viability at 0.1, 0.5, 1, 4, 6 h; Increased RH from low to mid levels associated with similar viability at 0, 23 h Mid to high RH: Increased RH from mid to high levels associated with similar viability | Low to mid RH (& low to high RH): Increased RH from low to mid levels associated with decreased viability at 0.1, 0.5, 1, 4, 6 h; Increased RH from low to mid levels associated with similar viability at 0, 23 h Mid to high RH: Increased RH from mid to high levels associated with similar viability Minimum viability associated with mid and high RH Maximum viability associated with low RH |
| Schaffer 1976 [40] USA | Influenza A (WSN_H strain) was aerosolized in a Wells refluxing atomizer (stirred settling chamber) | Humidity vs survival | Influenza A (WSN_H strain) | + | - | / | Allantoic Fluid | 1, 15, 30, 60 min | 21˚C | Increased RH from low to mid RH associated with decreased survival; increased RH from mid to high RH associated with relatively higher survival than at mid RH. | Increased RH from low to mid levels associated with decreased survival Increased RH from mid to high levels associated with increased survival Minimum survival associated with mid RH (40–60%RH) Maximum survival associated with low RH (<40%RH) |

(Continued)

**Table 1.** (Continued)

| First author Year Country | Experimental design | Outcome | Virus | Effect of Humidity | | | Medium | Exposure Times | Temperature | Result | Association |
|---|---|---|---|---|---|---|---|---|---|---|---|
| | | | | Low | Mid | High | | | | | |
| Noti 2013 [16] USA | Aerosolized Influenza A (H1N1) was coughed into a simulated examination room chamber using two manikins | Humidity vs infectivity | Influenza A (H1N1) | + | - | * | Hank's Balanced Salt Solution with 0.2% bovine serum albumin, 100 units/ml penicillin G, and 100 units/ml streptomycin (Blanchere [52]) | 5 coughs at 1 min intervals over 6 min | 20°C | Increased RH from 23% to 43% associated with decreased % infectivity (77.2% to 14.6%); Increased RH from 43% to 73% associated with similar % infectivity (14.6% to ~17%; Fig 3 in Noti 2013 [16]) Increased RH from 20% to 45% associated with *significant* decrease in infectious virus (Fig 4 in Noti 2013 [16]) | Increased RH from low to mid levels associated with *significantly* decreased infectivity Increased RH from mid to high levels associated with similar infectivity Minimum infectivity associated with mid and high RH (43–73% RH) Maximum infectivity associated with low RH (23%RH) |
| Van Doremalen 2013 [37] USA | Influenza A was aerosolized in an environmental chamber | Humidity vs virus viability | Influenza A [A/Mexico/ 4018/2009 (H1N1)] | NR | * | | Dulbecco's Modified Eagle Medium | Continuous sampling during aerosolization | 20°C | Increased RH from 40% to 70% had *no significant* effect on viability | Increased RH from mid to high levels had *no significant effect* on viability Low RH was not reported RH % for minimum and maximum viability not determined (not statistically significant) |
| Kormuth 2018 [41] USA | Influenza A (H1N1) was aerosolized into a rotating drum | Humidity vs infectivity | Influenza A (H1N1) | * | * | * | Human Bronchial Epithelial Extracellular Material (HBE ECM) | 35 min, 1 hr | 25 ± 1°C | RH had *no significant* effect on infectivity of H1N1 in HBE ECM | Increased RH from low to mid levels and increased RH from mid to high levels associated with *no significant* effect RH % for minimum and maximum decay not determined (not statistically significant) |

Influenza A WSN$_H$ strain [40], H1N1 [16, 41], and the Influenza A/Mexico/4018/2009 (H1N1) [37]. Settings included a 4 m$^3$ room [18, 41], rotating drums [33, 34], a Wells refluxing atomizer or stirred settling chamber [16], and environmental chamber [37].

Three studies found that increased humidity from low to mid RH associated with decreased virus survival [18, 40] and infectivity [16]. Harper [33, 34] found increasing humidity from low to mid RH was associated with decreased viability at 7–8°C at exposure times of 0, 0.1, 4, 6, and 23 hours, 20.5–24°C and exposure times of 0.1, 0.5, 1, 4, 6, and 23 hours, and at 32°C with exposure times of 0.1, 0.5, 1, 4, and 6 hours. Harper [33, 34] found that increasing humidity from low to mid RH was associated with similar viability at 7–8°C with exposure times of 0.5 and 1 hour, at 20.5–24°C with an exposure of 0 hours, and at 32°C with exposure times of 0 and 23 hours. When increasing humidity from low to mid RH, Kormuth et al. [41] found no significant effect of humidity on infectivity; Van Doremalen et al. [37] did not test low RH levels.

For increased humidity from mid to high RH, two studies found decreased survival [18] and significantly decreased infectivity [16]. Harper [33, 34] found increased humidity from mid to high RH was associated with decreased viability at 7–8°C and an exposure time of 23 hours. One study found that increased humidity from mid to high RH was associated with increased survival [40]. Harper [33, 34] found increased viability when increasing humidity from mid to high RH at 7–8°C and an exposure time of 0 and 0.1 hours. Two studies found no significant effect when increasing humidity from mid to high RH [37, 41]. As well, Harper [33, 34] found similar viability when increasing humidity from mid to high RH at 20.5–24°C and 32°C at all exposure times and at 0.5, 1, 4, and 6 hours for 7–8°C.

Unlike coronaviruses, many of the influenza studies presented consistent results for minimum and maximum survival versus humidity level. Most consistently, four of the six aerosol influenza studies found that low RH was associated with maximum survival (~15%RH and <40%RH, respectively) [18, 40], viability (<40%RH) [33, 34], and infectivity (23%RH) [16]. For minimum survival, Hemmes et al. [18] found that high RH was associated with minimum survival (~90%RH) and Harper [33, 34] found that high RH was associated minimum viability

**Table 2. Modelling studies.**

| First author Year Country | Study Design | Virus | Humidity level tested | Outcomes | Association |
|---|---|---|---|---|---|
| **Coronaviruses** | | | | | |
| Spena 2020 [42] Italy | Experimental data from literature was used to develop a model to determine the influence of humidity on SARS-CoV-2 viral survival load | SARS-CoV-2 | ASHRAE comfort zone "for domestic and office-like environments" (p.4) [four corners on psychrometric chart] 1. 80%RH; 20˚C 2. 50%RH; 26˚C 3. 30%RH; 20˚C 4. 20%RH; 27˚C | Viral Survival Load at 1-hour v specific enthalpy ". . .optimal pairs of temperature and relative humidity values for coronavirus viral load inactivation, wherein SARS-CoV-2 infectivity actually appears to be nearly suppressed." (p.9) | Optimal pairs [three corners on psychrometric chart] 1. 80%RH; 20˚C 2. 50%RH; 26˚C 3. 45%RH; 26˚C High and mid RH optimal pairs associated with coronavirus inactivation |
| **Influenza** | | | | | |
| Zuk 2009 [43] Poland | A heuristic model of Influenza A transmission was developed using experimental results of Lowen et al. (2007) to determine transmission as a function of temperature and relative humidity | Influenza A | 20%, 35%, 50%, 65%, and 80% | gamma vs RH, transmission vs RH | At 5˚C Increased RH from 35% to 80% associated with lower transmission rates Increased RH from low to mid associated with decreased transmission Increased RH from mid to high associated with decreased transmission |
| Posada 2010 [44] USA | A mathematical model using mathematical exponential decay was used to predict the viability of Influenza A using data from Schaffer et al. (1976) as a function of humidity | Influenza A | 20%-80% | Viability vs RH | Increased RH from low to mid levels associated with decreased viability Increased RH from mid to high levels associated with increased viability |
| Yang 2011 [17] USA | The size distribution and dynamics of Influenza A viruses emitted from a cough in typical residential and public settings was modeled over a large relative humidity range using data from Harper (1961) | Influenza A | 10% - 90% | IAV inactivation rate, concentration, distribution, and removal efficiency vs. RH and two different ACH. IAV size distribution and removal efficiency at fixed RH and two different ACH | Increased RH from 10% to 50% associated with decreased virus concentration; increased RH from 50% to 90% associated with decreased virus concentration Increased RH from low to mid levels associated with decreased virus concentration Increased RH from mid to high levels associated with decreased virus concentration |
| Halloran 2012 [45] USA | A Gaussian breath plume model for expiratory aerosols was used to determine the effect of relative humidity on transmission of Influenza virus using conditions similar to those used by Lowen et al. (2007) | Influenza | 0% - 100% | Virus Transmission vs. Ventilation/RH | For RH from 0% to 80%RH. Similar probability for RH from 20% to 80% For RH <80%RH Probability decreased at >95%RH at 20C and 30C for pulmonary; Probability decreased at >85%RH at 5C for pulmonary; Probability increased at >95%RH at 5C, 20C and 30C for nasopharyngeal-tracheobronchial Decreasing temperature from 20 to 5 associated with increased probability Increasing temperature from 20 to 30 associated with decreased probability |

*(Continued)*

**Table 2.** (Continued)

| First author Year Country | Study Design | Virus | Humidity level tested | Outcomes | Association |
|---|---|---|---|---|---|
| Koep 2013 [46] USA | Using field measurements from two Minnesota grade schools and five published animal studies, a Auto-Regressive Conditional Heteroskedasticity model was used to determine the effect of humidity in the reduction of influenza virus survival | Influenza | 2.64–9.45 mb AH | Influenza survival vs. AH | Increased AH from 2.67 mb to 9.45 mb AH associated with decreased influenza virus survival (75% to 45% survival) |
| | | | 40 and 60% RH | Influenza survival vs. RH | Increased RH from 40%to 60% associated with decreased influenza survival [~47% (Fig 4 in Koep 2013 [46]) to 34% survival (p.4)] |

at 7–8˚C (>60%RH). Both Noti et al. [16] and Harper [33, 34] at 20.5–24˚C and 32˚C found that both mid and high RH was associated with minimum virus survival and infectivity as there was little to no difference in survival and infectivity at the two RH ranges (43–73%RH and >40%RH, respectively). Schaffer et al. [40] found that mid RH was associated with maximum survival (40–60%RH). Two studies found that humidity was not associated with any significant difference in infectivity [41] and viability [37], as such, minimum and maximum survival could not be determined.

## Modelling studies

**Coronaviruses.** One modelling study [42] examined the effect of humidity on SARS-CoV-2 in terms of viral survival load per hour to determine optimal temperature/RH pairs for virus inactivation (Table 2). Spena et al. [42] used experimental data from Pyankov et al. [38] and Van Doremalen et al. [37] for MERS-CoV, MERS isolate HCov-EMC, SARS-CoV-1, and SARS-CoV-2 in the development of the model. Spena et al. [42] noted that specific enthalpy is a better predictor of ideal virus control than humidity; their study indicates a target value of 55kJ/kg is optimal. Unfortunately, this target results in high absolute humidity values well above typical set points for mechanical systems. To achieve 55kJ/kg specific enthalpy, HVAC settings require an indoor RH of approximately 93% at 20˚C, decreasing almost linearly to 60%RH at 25˚C (Fig 5 in Spena et al. [42]). Spena et al. [42] indicate a triangular subsector on the psychrometric chart within the ASHRAE recommended quadrangular comfort zone which is both optimal for virus control and comfort. Their findings indicate an important trade-off exists between controlling virus activity and typical building indoor air design parameters.

**Influenza.** Five modelling studies examined the effect of influenza (Table 2) [17, 43–46]. Three of the five studies [43, 45, 46] used animal transmission data from Lowen et al. [14] and one study [17] included data from Harper [33] for aerosolized viruses. Model types included a heuristic model [43], a mathematical model using mathematical exponential decay [44], a Gaussian breath plume model [45], an Auto-Regressive Conditional Heteroskedasticity model [46], and a model for the size and dynamics of Influenza A [17].

Three studies found that increased humidity from low to mid RH was associated with decreased virus transmission [43], viability [44], and concentration [17]. Two studies found that increased humidity from mid to high RH was associated with decreased virus transmission [43] and virus concentration [17]. One study found that increased humidity from mid to high RH was associated with increased viability [44]. Koep et al. [46] found that increased AH from 2.67 mb to 9.45 mb AH was associated with decreased virus survival and that increased

**Table 3. Experimental animal studies.**

| First author Year Country | Experimental Summary | Outcome | Virus | Effect at each RH* | | | Temperature | Data | Association |
|---|---|---|---|---|---|---|---|---|---|
| | | | | Low | Mid | High | | | |
| Loosli 1943 [47] USA | Groups of 10 mice were placed in a room with aerosolized Influenza for 20+ minutes at varying RHs (17–90%) | Humidity vs virus persistence (determined by infections in exposed mice) | Influenza A (PR8) | + | / | - | 27–29°C | Increased RH from 23% to 48% to 89% associated with decreased persistence of Influenza over time | Increased RH from low to mid levels associated with decreased persistence at 27–29°C Increased RH from mid to high levels associated with decreased persistence at 27–29°C Minimum viability associated with high RH (89%RH) Maximum viability associated with low RH (23–43%RH) |
| Lester 1948 [48] USA | Naive mice in groups of 10 were placed in wire cages divided into compartments in a room and exposed to aerosolized Influenza A | Humidity vs infectivity (determined by fatalities) | Influenza A (PR8) | + | - | + | 72–75°F (22.2–23.8°C) | Increased RH from 23% to 60% RH associated with decreased fatalities (100% to 22.5%); Increased RH from 60% to 80% associated with increased fatalities (22.5% to 100%) | Increased RH from low to mid levels associated with decreased infectivity (decreased fatalities) at 22.2–23.8°C Increased RH from mid to high levels associated with increased infectivity (increased fatalities) at 22.2–23.8°C Minimum viability associated with mid RH (45–60%RH) Maximum viability associated with low and high RH (23% and 80%RH) |
| Lowen 2007 [14] USA | Inoculated and naive guinea pigs were housed in adjacent cages inside an environmental chamber | Humidity vs transmission | Influenza A [Influenza A/ Panama/2007/99 (Pan/99; H3N2)] | + | / | - | 20°C | Increased RH from 20% to 50% associated with decreased transmission (100%, 75% to 25%,25%); Increased RH from 50% to 80% associated with decreased transmission (25%, 25% to 0%, 0%) | Increased RH from low to mid levels associated with decreased transmission at 5°C and 20°C Increased RH from mid to high levels associated with increased transmission at 5°C and 20°C Minimum viability associated with high RH (80%RH) Maximum viability associated with low RH (20%RH) |
| | | | | + | / | - | 5°C | Increased RH from 35% to 50% associated with a little change in influenza transmission (100%, 100% to 100%, 75%); Increased RH from 50% to 80% associated with decreased transmission (100%, 75% to 50%. 50%) | |

(*Continued*)

**Table 3.** (Continued)

| First author Year Country | Experimental Summary | Outcome | Virus | Effect at each RH* | | | Temperature | Data | Association |
|---|---|---|---|---|---|---|---|---|---|
| | | | | Low | Mid | High | | | |
| Steel 2011 [50] USA | Inoculated and naive guinea pigs were housed in adjacent cages inside an environmental chamber | Humidity vs transmission | Influenza A/ Panama/2007/ 1999 (H3N2) (Pan/99) | + | NR | - | 20°C | Increased RH from 20% to 80% associated with decreased transmission (100%, 100%, 75% to 25%, 0%, 0%) | Increased RH from low to high associated with decreased transmission 20°C and 30°C Mid RH not reported Minimum viability associated with high RH (80%RH) Maximum viability associated with low RH (20%RH) |
| | | | | + | NR | - | 30°C | Increased RH from 20% to 80% associated with decreased transmission (25%, 0% to 0%, 0%) | |
| | | | Influenza A/ Netherlands/602/ 2009 (H1N1) (NL/ 09) | + | NR | - | 20°C | Increased RH from 20% to 80% associated with decreased transmission (100% to 0%) | |
| | | | | + | NR | - | 30°C | Increased RH from 20% to 80% associated with decreased transmission (25% to 0%) | |
| Lowen 2014 [15] USA | Inoculated and naive guinea pigs were housed in adjacent cages inside an environmental chamber | Humidity vs transmission | Influenza A A/ Panama/2007/ 1999 (H3N2) and A/Netherlands/ 602/2009 (H1N1) | + | NR | NR | 5°C | Previously unpublished data: 100% transmission at 5°C and 20% RH Increased RH from 20% to 50% associated with a little change in influenza transmission (100% to 100%, 75%) where 50% RH data is from Lowen et al. (2007) | Increased RH from low to mid associated with decreased transmission at 5°C Mid RH data from Lowen et al. (2007) Maximum viability associated with low RH (20%RH) |
| Gustin 2015 [49] USA | Inoculated and naive ferrets were housed in adjacent cages inside an environmental chamber | Humidity vs transmission | Influenza A/ Panama/2007/ 1999 (H3N2) | + | - | + | 23°C | Increased RH from 30% to 50% associated with decreased influenza transmission (2/3 to 1/3); Increased RH from 50% to 70% associated with increased transmission (1/ 3 to 2/3) | Increased RH from low to mid associated with decreased transmission at 23°C Increased RH from mid to high associated with associated with increased transmission at 23°C Minimum viability associated with mid RH (50%RH) Maximum viability associated with low RH (30%) |
| | | | Influenza A/ Indiana/8/2011 (H3N2v) | + | - | / | 23°C | Increased RH from 30% to 50% associated with decreased influenza transmission (3/3 to 0/3); Increased RH from 50% to 70% associated with increased influenza transmission (0/3 to 2/3) | |

RH from 40% to 60%RH was associated with decreased survival. Halloran et al. [45] found humidity from 20% to 80%RH was associated with similar transmission probability. Additionally, Halloran et al. [45] found that decreasing temperature from 20°C to 5°C was associated with increased transmission probability, while increasing temperature from 20°C to 30°C was associated with decreased transmission probability.

## Animal studies

**Influenza.** Six animal studies examining the effect of humidity on viruses used influenza (Table 3) [14, 15, 47–50]. Three studies came from the same research group [14, 15, 50].

Strains used were Influenza A (PR8) [47, 48], Influenza A/Panama/2007/99 (Pan/99; H3N2) [14], Influenza A/Panama/2007/1999 (H3N2) (Pan/99) [15, 49, 50], Influenza A/Netherlands/602/2009 (H1N1) (NL/09) [15, 50], and Influenza A/Indiana/8/2011 (H3N2v) [49].

Five studies found that increased humidity from low to mid RH was associated with decreased virus persistence [47], infectivity [48], and transmission [14, 15, 49].

Two studies found that increased humidity from mid to high RH was associated with decreased virus persistence at 5˚C, 20˚C, and 27–29˚C [14, 47]. However, three other studies found that increased humidity from mid to high RH was associated with increased infectivity at 22.2–23.8˚C [48] and transmission at 20˚C (65%RH) and 23˚C [14, 49]. Steel et al. [50] found that increased humidity from low to high RH (mid RH not tested) was associated with decreased transmission at 20˚C and 30˚C.

Like that of aerosolized influenza studies, six studies found that low RH was associated with maximum virus survival (23–43%RH [47], 23%RH [48], 20%RH [14], 20%RH [15], 20%RH [50], and 30%RH [49], respectively). Three studies found that high RH (89%RH [47], 80%RH [14], and 80%RH [50]) was associated with minimum virus survival and two studies found mid RH was associated with minimum virus survival (45–60%RH [48] and 50%RH [49], respectively).

## Field studies

**Influenza.** One study found no significant effect of absolute and relative humidity on Influenza A and B detection in different settings on a university campus in Hong Kong (Table 4) [51].

Table 5 shows a visual representation of the relative change (↑ increase, ↓ decrease,—no change) in virus infectivity between low (<40%), mid (40%-60%), and high (>60%) RH.

## Risk of bias

All animal and field experimental studies had low risk of bias for the three domains: selection bias, information bias and confounding. Seven of the aerosolized virus experimental studies had low risk of bias for all three domains. For the remaining aerosolized virus experimental studies, one had unclear information bias due to lack of clarity regarding exposure time [18] and one had unclear information bias and high selection bias because the test and tracer material were not identical [33, 34]. One was assessed with high risk of bias due to confounding for our comparison of interest because both humidity and temperature were changed, where 79% RH and 25˚C was compared with 24%RH and 38˚C [38]. The six modelling studies had low risk of bias for all three domains: definition, assumption, and validation.

**Table 4. Field studies.**

| First author Year Country | Setting/Population | Study Type | Humidity level tested | | Investigated Parameter | Result |
|---|---|---|---|---|---|---|
| | | | AH | RH | | |
| Xie 2020 [51] China | University campus in Hong Kong. Locations include canteens, lecture halls, shuttle buses, and the University Health Service | Observational | 4.2–22.9 g/m³ | 27.1%–98.3% | Effect of absolute humidity and relative humidity on Influenza A and B detection in air | AH did not have a statistically significant association with influenza detection; RH included in univariate analysis (P value = 0.752) but not multivariate analysis. |

**Table 5. Virus infectivity relative change between low RH (<40%RH), mid RH (40%-60%RH), high RH (>60%RH).**

| Study | Virus | Low to mid RH | Mid to high RH | Low to high RH |
|---|---|---|---|---|
| **Temperature ~ 20°C** | | | | |
| **Coronavirus** | | | | |
| *Aerosolized* | | | | |
| Ijaz 1985 [39] | hCoV-229E | ↑ | ↓ | |
| van Doremalen 2013 [37] | MERS-CoV | | ↓ | |
| Pyankov 2018 [38] | MERS-CoV | | | ↑ (38 C to 25 C) |
| Smither 2020 [36] | SARS-CoV-2 in Tissue Culture Medium | | ↑ | |
| Smither 2020 [36] | SARS-CoV-2 in Artificial Saliva | | ↑ 15, 30, 90 min / - 0, 60 min | |
| Schuit 2020 [35] | SARS-CoV-2 | - | - | |
| **Influenza** | | | | |
| *Aerosolized* | | | | |
| Hemmes 1960 [18] | Influenza A (PR8) | ↓ | ↓ | |
| Harper 1961/1963 [33, 34] | Influenza A | ↓ 0.1,0.5,1,4,6,23 h / - 0 h | - | |
| Schaffer 1976 [40] | Influenza A (WSN$_H$) | ↓ | ↑ | |
| Noti 2013 [16] | Influenza A (H1N1) | ↓ | - | |
| van Doremalen 2013 [37] | Influenza A (H1N1) | | - | |
| Kormuth 2018 [41] | Influenza A (H1N1) | - | - | |
| *Animal* | | | | |
| Loosli 1943 [47] | Influenza A (PR8) | ↓ | ↓ | |
| Lester 1948 [48] | Influenza A (PR8) | ↓ | ↑ | |
| Lowen 2007 [14] | Influenza A (H3N2) | ↓ | ↑ 65%RH / ↓ 80%RH | |
| Steel 2011 [50] | Influenza A (H3N2) | | | ↓ |
| Steel 2011 [50] | Influenza A (H1N1) | | | ↓ |
| Gustin 2015 [49] | Influenza A (H3N2) | ↓ | ↑ | |
| Gustin 2015 [49] | Influenza A (H3N2v) | ↓ | ↑ | |
| **Temperature ~5°C** | | | | |
| **Coronavirus** | | | | |
| *Aerosolized* | | | | |
| Ijaz 1985 [39] | hCoV-229E | ↑ | ↓ | |
| **Influenza** | | | | |
| *Aerosolized* | | | | |
| Harper 1961/1963 [33, 34] | Influenza A | ↓ 0,0.1,4,6,23 h / - 0.5,1 h | ↑ 0,0.1 h / - 0.5,1,4,6, h / ↓ 23 h | |
| *Animal* | | | | |
| Lowen 2007 [14] 2014 [15] | Influenza A (H3N2) | ↓ | ↓ | |
| **Temperature ~30°C** | | | | |
| **Influenza** | | | | |
| *Aerosolized* | | | | |
| Harper 1961/1963 [33, 34] | Influenza A | ↓ 0.1,0.5,1,4,6 h / - 0,23 h | - | |
| *Animal* | | | | |
| Steel 2011 [50] | Influenza A (H3N2) | | | ↓ |
| Steel 2011 [50] | Influenza A (H1N1) | | | ↓ |

## Discussion

This systematic review focussed on the HVAC design feature of humidity and its effect on transmission of coronavirus and influenza, both enveloped, ssRNA viruses. Several important findings were revealed. First, increased humidity from mid to high RH for SARS-CoV-2 was associated with decreased virus survival. Second, although SARS-CoV-2 results appear consistent, coronaviruses do not all behave the same and consistent minimum/maximum survival versus humidity level could not be determined. Third, increased humidity from low to mid RH for influenza was associated with decreased persistence, infectivity, viability, and survival; however, increased humidity from mid to high RH for influenza did not show consistent results. Fourth, low humidity was associated with maximum influenza survival; however, the humidity level for minimum survival was not consistent. Fifth, even though both were enveloped, ssRNA viruses, coronaviruses and influenza did not behave the same. For example, SARS-CoV-2 data found that increased humidity from low to mid RH or mid to high RH was associated with either no effect [35] or increased survival [36], while influenza data, using H1N1 as an example, found that increased humidity from mid to high RH was associated with either no effect [16, 37, 41] or decreased transmission [50]. Sixth, similar to results reported in previous reviews [12, 13] of humidity and viruses, medium, temperature, and exposure time contributed to inconsistency in results for both coronaviruses and influenza.

The relationship between airborne virus transmission and relative humidity is decidedly complex. Ijaz et al. [39] propose that airborne survival of vertebrate viruses under various environmental conditions cannot be predicted based on viral structure and composition. According to Lowen et al. [15], there is likely more than one mechanism by which relative humidity affects virus transmission. Not only does relative humidity affect viral particles, it can also have an impact on the host. Lowen et al. [15] suggest that low relative humidity can damage nasal epithelia and reduce mucociliary clearance. This would render the host more susceptible to respiratory virus infections.

Temperature and suspending medium are oftentimes entangled with the effects of relative humidity [33, 34, 39, 50]. However, Hemmes et al. [18] asserted that relative humidity has a larger effect on the survival of aerosolized viruses compared to temperature. Salt and protein concentrations in the suspending medium can have a marked effect on the aerosol stability of a virus [37]. Lester [48] found that decreasing the salt concentration of influenza A virus-lung suspension eliminated the deleterious effect of increasing the relative humidity to 50%. Kormuth et al. [41] found that human bronchial epithelial extracellular material (HBE ECM) protected aerosolized influenza virus from relative humidity dependent decay. They go on to state that protein is most likely protecting the virus from decay but other elements of HBE ECM should not be ruled out.

Relative humidity also affects the settling of virus-containing respiratory droplets. High relative humidity is linked with increased settling [14, 17, 41], thereby preventing the formation of droplet nuclei [14]. However, Yang and Marr's [17] analysis revealed that relative humidity plays a larger role in virus inactivation than removal through settling.

It is understood that increasing humidification is not feasible in all types of facilities due to existing design limitations [16, 46]. Noti et al. [16] suggest that high risk, low humidity areas should be identified during the design and construction phase and appropriate consideration should be given to designs that minimize infection risk. Spena et al. [42] identified a region on the psychrometric chart that satisfies ASHRAE Standard 55's comfort zone requirements while also providing optimal humidity conditions to decrease SARS-CoV-2 survival. In general, Spena et al. [42] suggest increasing humidification of supply air in the winter season and decreasing dehumidification of supply air in the summer season.

## Implications for research

Based on the included studies, several key implications for research were found, such as the influence of medium, temperature, and exposure time; the need for statistical analysis to better understand and interpret results; and the need for standardized testing procedures. For example, for SARS-CoV-2, increased humidity from low to high RH was associated with an increase or no change in infectivity, where the difference was attributed to differences in exposure time and/or suspending medium. Interestingly, increased humidity from low to high RH was associated with an increase in infectivity for MERS-CoV, where one study decreased the temperature while increasing the RH. While this is one example, the complexity of other factors can be seen through the review content.

While one study [36] directly compared mediums, it was not the only study to comment on the perceived influence of medium on the results. Smither et al. [36] compared Artificial Saliva (AS) and tissue culture medium (TCM), finding that there were fewer particles observed in AS than in TCM, perhaps contributing to the amount of viable virus present. While the difference in medium has implications for research, practitioners may want to consider the results from AS as they are more applicable to real-world transmission or virus survival scenarios. Additionally, using human bronchial epithelial extracellular material (HBE ECM), Kormuth et al. [41] found no significant effect of humidity on viruses. Kormuth et al. suggested that a lack of results could be due to "protection conferred by supplementation of the viruses with HBE ECM" [41 (p744)]. This theory was further tested using Phi6 both with and without HBE ECM. Additionally, Kormuth et al. questioned "how well the media composition represented that of actual aerosols and droplets expelled by an infected host" [41 (p744)]. As such, researchers should be aware of the influence of medium when testing humidity while also considering their research goals (e.g., choosing a medium such as AS to simulate real-world scenarios).

Five studies used multiple temperatures when examining the influence of humidity on viruses [14, 15, 33, 34, 39, 50]. The relationship between temperature and humidity is complex [42]. For example, Ijaz et al. found that "the fluidity of the lipid-containing envelope is stabilized at low temperature, thus protecting the virion; however, further studies are needed to explain these phenomena" [39 (p2747)]. As well, the Lowen group [14, 15, 50] found that humidity and temperature as a combined approach could have an impact on virus survival, suggesting that "influenza virus transmission indoors could potentially be curtailed by simply maintaining room air at warm temperatures (20˚C) and either intermediate (50%) or high (80%) RHs" [14 (p1475)]. Additionally, Harper [33, 34] found that high temperatures were associated with the lowest survival at all levels of humidity, finding the influence of humidity "negligible" [34 (p68)]. As a result, researchers should consider the influence of temperature in addition to the influence of humidity.

Exposure time was also found to be a factor when testing the effect of humidity [42]. Schuit et al. [35] questioned whether insufficient exposure time explained the inability to detect a RH relationship for SARS-CoV-2 that was similar to hCoV-229E. Smither et al. [36] found that, in conjunction with the effect of medium, increased exposure time resulted in two patterns of virus survival. For SARS-CoV-2 in AS, increased exposure time was associated with increased differences between virus survival between mid and high RH. In TCM, increased exposure time was associated with decreased differences between survival at mid and high RH. Similarly, Harper [33, 34] found that aerosols were able to remain viable for a considerable amount of time "in favourable conditions" [33 (p485)]. However, Harper [33] also noted that favourable conditions vary by virus.

Other research implications arising from this review are the need for reporting of statistical analyses and standard procedures for testing. Only four of the 18 experimental studies included statistical analysis in their presentation of results [16, 35, 37, 41]. Two of the four studies found significant results [16, 37]; while two studies found nonsignificant results which the authors attributed to study design issues [35, 41]. A limitation of this review is that studies with statistical analyses were presented alongside studies that did not conduct or report statistical analyses in order to allow for a full understanding of the available humidity/virus literature. While attempts were made to ensure clarity when reporting results, differences between survival were reported without supporting statistical analysis from the original documentation and as such, could potentially influence overall findings. Additionally, these findings are further complicated by inconsistencies in testing procedures across the included studies. For example, Schuit et al. [35] attributed their nonsignificant results to short exposure times and Kormuth et al. [41] suggested that medium choice may have affected study outcomes. Additionally, as Derby et al. [12] previously noted, not all of the studies tested a full spectrum of RH levels. Two experimental studies [36, 37] did not test low RH (<40%RH) [36, 37] and two experimental studies [38, 50] did not test mid RH (40–60%RH). As a result, it can be difficult to make comparisons even among similar viruses.

## Implications for practice

In their January 2021 Core Recommendations for Reducing Airborne Infectious Aerosol Exposure ASHRAE recommended "maintaining temperatures and humidity at set points" [11]. Spena et al. [42] considered the ASHRAE comfort zone for domestic and office-like environments in their model of SARS-CoV-2. They found that SARS-CoV-2 infectivity would be effectively suppressed for only a portion of the temperature and RH in the ASHRAE comfort zone. This indicated target zone is of high humidity and would be challenging in buildings where mould/mildew control is important, or in older buildings in cold climates where condensation can be a problem. Cold regions would require very significant humidification efforts when outdoor make-up air has very low absolute humidity at inlet. As this modelling study by Spena et al. [42] was released early in the SARS-CoV-2 pandemic timeline, the model includes data for Phi6, HCoV-EMC (MERS), SARS-CoV, and one SARS-CoV-2 study to provide inputs for SARS-CoV-2 modelling. While the enthalpy was not calculated within the scope of this review, the results of Schuit et al. [35] and Smither et al. [36] potentially do not support the findings of Spena et al. [42]. Schuit et al. [35] did not find a significant effect of humidity, possibly due to short exposure times. However, Smither et al. [36] found that high RH was associated with maximum survival at 19–22˚C, whereas the modelling by Spena et al. [42] suggests that mid to high RH is associated with increased inactivation or decreased survival. As such, it would be interesting to see how the results of Spena et al. [42] change with new research.

## Strengths and limitations

Comprehensiveness and use of methods to avoid bias, including an a priori protocol, pre-specification of inclusion/exclusion criteria and involvement of at least two reviewers at all stages, are the strengths of this study. The limitations of this study are the inconsistencies across the included studies regarding statistical analysis and standardized testing procedures. To gain a full understanding of the available literature, included studies with statistical analysis were presented alongside included studies without statistical analysis. Comparison of included studies of the same virus was challenging due to a lack of standardized testing procedures regarding exposure time, temperature, and medium.

## Future research and practice priorities

Even though increasing relative humidity from mid to high RH was associated with decreased SARS-CoV-2 survival, all coronaviruses do not behave the same. As a result, blanket prescriptive humidity levels for coronavirus mitigation are difficult to ascertain. While influenza survival varied from mid to high RH, increased humidity from low to mid RH was associated with decreased virus survival with maximum survival at low RH. When controlling humidity as an HVAC feature, practitioners should take into account virus type and temperature. Future research should also consider the impact of exposure time, temperature, and medium when designing experiments, while also working towards more standardized testing procedures and statistical analysis.

## Conclusions

This systematic review identified 24 studies examining the role of humidity as an HVAC intervention and its effect on transmission of the lipid enveloped, ssRNA influenza and coronaviruses. Similar to previous reviews [12, 13], it was found that while humidity can have an effect on viruses, aerosol medium, temperature, and exposure time can also influence the role of humidity. While SARS-CoV-2 results appear to be consistent as increased humidity from mid to high RH was associated with decreased virus survival, not all coronaviruses behave the same way. Additionally, increasing humidity from low to mid RH for influenza was associated with decreased survival; however, increasing humidity from mid to high RH for influenza was not consistent. When examining humidity as a HVAC intervention, medium, temperature, and exposure time should be considered. As well, due to inconsistencies across viruses, while recommended levels for specific viruses could potentially be determined, generalized approaches to humidity cannot be made.

## Supporting information

**S1 Checklist. PRISMA 2009 checklist.**
(DOC)

**S1 File.**
(DOCX)

## Acknowledgments

We thank Tara Landry and Alison Henry for conducting the peer review of the search strategies. We thank Samuel Ducholke, Kristen Rumbold, Larry Zhong, and Stella Mathews for their involvement in screening studies for inclusion, and Natalie Fleck for her involvement in screening studies for inclusion and assessing studies for risk of bias.

## Author Contributions

**Conceptualization:** Gail M. Thornton, Brian A. Fleck, Lexuan Zhong, Lisa Hartling.

**Data curation:** Gail M. Thornton, Brian A. Fleck, Dhyey Dandnayak, Emily Kroeker, Lisa Hartling.

**Formal analysis:** Gail M. Thornton, Brian A. Fleck, Dhyey Dandnayak, Emily Kroeker, Lisa Hartling.

**Funding acquisition:** Brian A. Fleck, Lexuan Zhong, Lisa Hartling.

**Investigation:** Gail M. Thornton, Brian A. Fleck, Dhyey Dandnayak, Emily Kroeker, Lisa Hartling.

**Methodology:** Gail M. Thornton, Brian A. Fleck, Lisa Hartling.

**Project administration:** Gail M. Thornton, Brian A. Fleck, Lisa Hartling.

**Resources:** Brian A. Fleck.

**Supervision:** Gail M. Thornton, Brian A. Fleck.

**Validation:** Gail M. Thornton, Brian A. Fleck.

**Visualization:** Gail M. Thornton, Brian A. Fleck, Dhyey Dandnayak, Emily Kroeker, Lisa Hartling.

**Writing – original draft:** Gail M. Thornton, Brian A. Fleck, Dhyey Dandnayak, Emily Kroeker, Lisa Hartling.

**Writing – review & editing:** Gail M. Thornton, Brian A. Fleck, Dhyey Dandnayak, Emily Kroeker, Lexuan Zhong, Lisa Hartling.

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
