## [Decision Letter · Decision Letter 0]

22 Aug 2022

PONE-D-22-01074

The impact of heating, ventilation and air conditioning (HVAC) design features on the transmission of viruses, including the 2019 novel coronavirus (COVID-19): a systematic review of humidity

PLOS ONE

Dear Dr. Fleck,

Thank you for submitting your manuscript to PLOS ONE. After careful consideration, we feel that it has merit but does not fully meet PLOS ONE’s publication criteria as it currently stands. Therefore, we invite you to submit a revised version of the manuscript that addresses the points raised during the review process.

The manuscript has been evaluated by three reviewers, and their comments are available below.

The reviewers have raised a number of minor concerns that need attention. Could you please revise the manuscript to carefully address the concerns raised?

We look forward to receiving your revised manuscript.

Kind regards,

Sebastian Shepherd

Staff Editor

PLOS ONE

“This work is funded by a Canadian Institutes of Health Research (CIHR) Operating Grant: Canadian 2019 Novel Coronavirus (COVID-19) Rapid Research Funding Opportunity [https://webapps.cihr-irsc.gc.ca/decisions/p/project_details.html?applId=422567&lang=en]. Dr. Hartling is supported by a Canada Research Chair in Knowledge Synthesis and Translation. Drs. Fleck and Zhong are supported by the NSERC Discovery program.”

“This work is funded by a Canadian Institutes of Health Research (CIHR) Operating Grant, Canadian 2019 Novel Coronavirus (COVID-19) Rapid Research Funding Opportunity [https://webapps.cihr-irsc.gc.ca/decisions/p/project_details.html?applId=422567&lang=en], and Alberta Innovates. Funding was received by LZ. The funders played no role in study design, data collection and analysis, decision to publish, or preparation of the manuscript.”

“BF is an unpaid advisor for Pura Air Inc. in Vancouver.”

4. Please upload a copy of Figures 3-5, to which you refer in your manuscript. If the figure is no longer to be included as part of the submission please remove all reference to it within the text.

Reviewers' comments:

Reviewer's Responses to Questions

**Comments to the Author**

1. Is the manuscript technically sound, and do the data support the conclusions?

Reviewer #1: Yes

Reviewer #2: Yes

2. Has the statistical analysis been performed appropriately and rigorously? 

Reviewer #1: N/A

Reviewer #2: N/A

3. Have the authors made all data underlying the findings in their manuscript fully available?

Reviewer #1: Yes

Reviewer #2: Yes

4. Is the manuscript presented in an intelligible fashion and written in standard English?

Reviewer #1: Yes

Reviewer #2: Yes

5. Review Comments to the Author

Reviewer #1: General comments:

This study performs a systematic literature review on the effects of humidity on the viability inside HVAC systems for different viruses. The authors concluded that other parameters, including aerosol medium temperature and exposure time, influence the effect of humidity on virus viability and found that humidity effects on viability are not the same for all viruses, and consideration must be taken for each virus type. I believe the manuscript is well written and an important and useful topic for researchers interested in laboratory and field studies. Therefore, I accept the manuscript with a few changes:

1) You have identified low, mid, and high levels of RH, but instead of using these definitions on and off, I would consistently use the RH values.

2) Please correct the in-text citation when using an author's name. For example, in line 63, Derby et al., please add the reference number. Please review the manuscript and correct all similar instances.

3) Please add two separate sections that address a) the limitations of this study, b) expand on "identify gaps in current research, which can guide future research priorities".

Reviewer #2: The authors systematically reviewed the effect of humidity on the survival and transmission of coronaviruses and influenza viruses. The work followed rigorous methods for study selection. It captured all the studies I would expect. The results are explained clearly, and the discussion effectively synthesizes the findings and relates them to the real world. I have just a few minor suggestions.

1. line 135: “A variety of agents were included in the search with priority placed on studies of viruses or agents that simulated viruses. Other agents (e.g., fungi, bacteria) would be included if studies were not available specific to viruses. Studies using bacteriophages, which are viruses that infect bacterial cells [26], were included.” This seems to contradict the objective stated in the introduction, which was to focus on “enveloped, ssRNA coronaviruses and influenza viruses as opposed to viruses more generally.”

2. line 232: “Influenza A/Mexico/4018/2019” I think this should be 2009 instead of 2019.

3. line 265: Consider pointing out that the modeling studies rely on experimental data, so their results are not independent of those presented in the previous subsection.

6. PLOS authors have the option to publish the peer review history of their article (what does this mean?). If published, this will include your full peer review and any attached files.

Reviewer #1: No

Reviewer #2: No

---

## [Author Response · Author response to Decision Letter 0]

6 Sep 2022

1. Please ensure that your manuscript meets PLOS ONE's style requirements, including those for file naming. The PLOSONE style templates can be found at

Authors’ response: We have reviewed the style template documents and made the required formatting changes.

“This work is funded by a Canadian Institutes of Health Research (CIHR) Operating Grant: Canadian 2019 Novel Coronavirus (COVID-19) Rapid Research Funding Opportunity [https://webapps.cihr-irsc.gc.ca/decisions/p/project_details.html?applId=422567&lang=en]. Dr. Hartling is supported by a Canada Research Chair in Knowledge Synthesis and Translation. Drs. Fleck and Zhong are supported by the NSERC Discovery program.”

Please remove any funding-related text from the manuscript and let us know how you would like to update your 

Funding Statement. Currently, your Funding Statement reads as follows:

“This work is funded by a Canadian Institutes of Health Research (CIHR) Operating Grant, Canadian 2019 NovelCoronavirus (COVID-19) Rapid Research Funding Opportunity [https://webapps.cihr-irsc.gc.ca/decisions/p/project_details.html?applId=422567&lang=en], and Alberta Innovates. Funding was received by LZ. The funders played no role in study design, data collection and analysis, decision to publish, or preparation of the manuscript.”

Authors’ response: We would like the following funding statement to appear in the online submission; thank you for making this change on our behalf: This work is funded by a Canadian Institutes of Health Research (CIHR) Operating Grant, Canadian 2019 Novel Coronavirus (COVID-19) Rapid Research Funding Opportunity [https://webapps.cihr-irsc.gc.ca/decisions/p/project_details.html?applId=422567&lang=en], and Alberta Innovates. Dr. Hartling is supported by a Canada Research Chair in Knowledge Synthesis and Translation. Drs. Fleck and Zhong are supported by the NSERC Discovery program. The funders played no role in study design, data collection and analysis, decision to publish, or preparation of the manuscript.

“BF is an unpaid advisor for Pura Air Inc. in Vancouver.”

Authors’ response: This is no longer current. BF is NOT an advisor for Pura Air Inc. We have removed this statement from the manuscript.

4. Please upload a copy of Figures 3-5, to which you refer in your manuscript. If the figure is no longer to be included as part of the submission please remove all reference to it within the text.

Authors’ response: These figures were from studies that we included in our systematic review; we are citing the source of the data that we have included from these relevant studies. In Table 1 (Page 15) in the row labelled Noti 201316, "Figure 3" and "Figure 4" refer to Figure 3 and Figure 4 in Noti 201316. Similarly, in Table 2 (Page 22) in the row labelled Koep 201346, "Figure 4" refers to Figure 4 in Koep 201346. On Line 273-274, “To achieve 55kJ/kg specific enthalpy, HVAC settings require an indoor RH of approximately 93% at 20°C, decreasing almost linearly to 60%RH at 25°C (Fig. 5 [42]).”; “Fig. 5” refers to Figure 5 in Spena et al [42].

In the revised manuscript we have clarified, e.g., Figure 3 in Noti 2013, Figure 4 in Noti 2013, etc. Is this satisfactory? If not, we could change the figure numbers to the page numbers in the cited sources. See below:

Line 277

To achieve 55kJ/kg specific enthalpy, HVAC settings require an indoor RH of approximately 93% at 20°C, decreasing almost linearly to 60%RH at 25°C (Fig. 5 in Spena et al [42]).

Could change to: To achieve 55kJ/kg specific enthalpy, HVAC settings require an indoor RH of approximately 93% at 20°C, decreasing almost linearly to 60%RH at 25°C [42p10].

Table 1: Noti 2013

Increased RH from 43% to 73% associated with similar % infectivity (14.6% to ~17%; Figure 3 in Noti 2013 [16])

Could change to: Increased RH from 43% to 73% associated with similar % infectivity (14.6% to ~17%; p.5)

Increased RH from 20% to 45% associated with significant decrease in infectious virus (Figure 4 in Noti 2013 [16])

Could change to: Increased RH from 20% to 45% associated with significant decrease in infectious virus (p.6)

Table 2: Koep 2013

Increased RH from 40% to 60% associated with decreased influenza survival [~47% (Figure 4 in Koel 2013 [46]) to 34% survival (p.4)]

Could change to: Increased RH from 40% to 60% associated with decreased influenza survival [~47% (p.6) to 34% survival (p.4)]

Authors’ response: We have reviewed the reference list and made minor edits to ensure the references are complete and correct. There were no retractions.

Reviewers' comments:

Reviewer #1: 

General comments:

This study performs a systematic literature review on the effects of humidity on the viability inside HVAC systems for different viruses. The authors concluded that other parameters, including aerosol medium temperature and exposure time, influence the effect of humidity on virus viability and found that humidity effects on viability are not the same for all viruses, and consideration must be taken for each virus type. I believe the manuscript is well written and an important and useful topic for researchers interested in laboratory and field studies. Therefore, I accept the manuscript with a few changes:

1) You have identified low, mid, and high levels of RH, but instead of using these definitions on and off, I would consistently use the RH values.

Authors’ response: We recognize that we were inconsistent in the use of low, mid, and high RH in the text. We have corrected these inconsistencies throughout the manuscript (please see Tracked Changes). Now, we consistently use various forms of low RH, mid RH, high RH, low to mid RH, mid to high RH, and low to high RH. In addition, we added the humidity categorization in the beginning of the Results section on Line 188-190: “Details of individual studies are provided in tables and summarized in the sections that follow; humidity was categorized as low (<40%RH), mid (40-60%RH), and high (>60%RH) RH.” We feel that the use of low, mid, and high RH is easier for the reader to follow than repeating the RH range in every instance, and also allows for synthesis across studies that used slightly different values.

2) Please correct the in-text citation when using an author's name. For example, in line 63, Derby et al., please add the reference number. Please review the manuscript and correct all similar instances.

Authors’ response: Where appropriate, we moved the reference number from the end of the sentence to appear after the author name for in-text citations throughout the manuscript (please see Tracked Changes). However, the example from Line 63 is different: “Derby et al reviewed the effect of low humidity (≤40% relative humidity [RH]) on virus viability and transmission [12] and identified several studies, both modelling and experimental, showing that humidity influenced virus transmission and virus survival [14-18].” Vancouver style indicates that “In-text citation numbers should be placed after the relevant part of a sentence.” (https://guides.hsict.library.utoronto.ca/vancouver-style). For the example from Line 63, different parts of the sentence refer to different citable information; therefore, the current citation formatting is retained.

3) Please add two separate sections that address a) the limitations of this study, b) expand on "identify gaps in current research, which can guide future research priorities".

Authors’ response: We added a “Strengths and Limitations” section on Line 473-481 and a “Future Research and Practice Priorities” section on Line 482-491.

Reviewer #2:

The authors systematically reviewed the effect of humidity on the survival and transmission of coronaviruses and influenza viruses. The work followed rigorous methods for study selection. It captured all the studies I would expect. The results are explained clearly, and the discussion effectively synthesizes the findings and relates them to the real world. I have just a few minor suggestions.

1. line 135: “A variety of agents were included in the search with priority placed on studies of viruses or agents that simulated viruses. Other agents (e.g., fungi, bacteria) would be included if studies were not available specific to viruses. Studies using bacteriophages, which are viruses that infect bacterial cells [26], were included.” This seems to contradict the objective stated in the introduction, which was to focus on “enveloped, ssRNA coronaviruses and influenza viruses as opposed to viruses more generally.”

Authors’ response: The following sentence has been added on Line 139: "For this specific review, the synthesis was further narrowed from viruses to coronaviruses and influenza viruses."

2. line 232: “Influenza A/Mexico/4018/2019” I think this should be 2009 instead of 2019.

Authors’ response: You are correct. Thank you for catching this error. The text now reads “Influenza A/Mexico/4018/2009” on Line 235 and “A/Mexico/4018/2009” in Table 1 on Page 16.

3. line 265: Consider pointing out that the modeling studies rely on experimental data, so their results are not independent of those presented in the previous subsection.

Authors’ response: We already carefully detail in the text the relationship between the modelling studies and the experimental data. Line 272-274 “Spena et al [42] used experimental data from Pyankov et al [38] and Van Doremalen et al [37] for MERS-CoV, MERS isolate HCov-EMC, SARS-CoV-1, and SARS-CoV-2 in the development of the model.” Line 285-287 “Three of the five studies [43,45,46] used animal transmission data from Lowen et al [14] and one study [17] included data from Harper [33] for aerosolized viruses.”

---

## [Decision Letter · Decision Letter 1]

21 Sep 2022

The impact of heating, ventilation and air conditioning (HVAC) design features on the transmission of viruses, including the 2019 novel coronavirus (COVID-19): a systematic review of humidity

PONE-D-22-01074R1

Dear Dr. Brian Fleck

We’re pleased to inform you thariuopt has been judged scientifically suitable for publication and will be formally accepted for publication once it meets all outstanding technical requirements.

Kind regards,

Jean-Luc EPH Darlix, MG, Ph.D.

Academic Editor

PLOS ONE

Additional Editor Comments (optional):

Reviewers' comments:

Reviewer's Responses to Questions

**Comments to the Author**

1. If the authors have adequately addressed your comments raised in a previous round of review and you feel that this manuscript is now acceptable for publication, you may indicate that here to bypass the “Comments to the Author” section, enter your conflict of interest statement in the “Confidential to Editor” section, and submit your "Accept" recommendation.

Reviewer #1: All comments have been addressed

Reviewer #2: All comments have been addressed

2. Is the manuscript technically sound, and do the data support the conclusions?

Reviewer #1: Yes

Reviewer #2: Yes

3. Has the statistical analysis been performed appropriately and rigorously? 

Reviewer #1: Yes

Reviewer #2: N/A

4. Have the authors made all data underlying the findings in their manuscript fully available?

Reviewer #1: (No Response)

Reviewer #2: Yes

5. Is the manuscript presented in an intelligible fashion and written in standard English?

Reviewer #1: Yes

Reviewer #2: Yes

6. Review Comments to the Author

Reviewer #1: The authors have addressed my comments. Therefore, no further editing is required, and I accept the manuscript for publication.

Reviewer #2: The authors have addressed the reviewers’ comments satisfactorily. I have a couple new comments.

1. Table 5: This table does not appear to be referenced in the text. The meaning of “relative change” in the caption should be clarified. I was surprised to see mostly red down arrows in the “low to mid RH” column. In most studies, these viruses survive better at low to mid RH than at higher RHs. The rightmost column should probably be titled “High RH” rather than “Low to high RH.” Carefully check this table.

2. line 356: “SARS-CoV-2 data found that increased humidity...” I suggest changing the wording, as data do not usually find; rather they can show or reveal.

7. PLOS authors have the option to publish the peer review history of their article (what does this mean?). If published, this will include your full peer review and any attached files.

Reviewer #1: No

Reviewer #2: **Yes: **Linsey Marr

---

## [Editor Report · Acceptance letter]

28 Sep 2022

PONE-D-22-01074R1 

The impact of heating, ventilation and air conditioning (HVAC) design features on the transmission of viruses, including the 2019 novel coronavirus (COVID-19): a systematic review of humidity 

Dear Dr. Fleck:

I'm pleased to inform you that your manuscript has been deemed suitable for publication in PLOS ONE. Congratulations! Your manuscript is now with our production department. 

Kind regards, 

on behalf of

Professor Jean-Luc EPH Darlix 

Academic Editor

PLOS ONE